# Fish Use of a Borland-Type Fish Lock in an Iberian River

Francisco N. Godinho [1],*, Paulo J. Pinheiro [1] and Liliana Benites [2]

1   AQUALOGUS, Engineering and Environment Lda, Rua do Mar da China 1, Office 2.4. Parque das Nações, 1990-137 Lisboa, Portugal
2   EDP—Gestão da Produção de Energia SA, Rua Ofélia Diogo Costa, 4250-468 Porto, Portugal
*   Correspondence: fgodinho@aqualogus.pt or franciscogodinho111@gmail.com

**Abstract:** Fish locks were fitted at dams to improve fish migration in Europe, but also in America and Australia. In Iberia, several dams were equipped with locks in the 1970s and 1980s, and in this study the fish use of the Borland fish lock installed in the most downstream dam in the Douro River was investigated by visualizing fish during each cycle. Moreover, a creel survey was conducted encompassing the same period to characterize the fish caught downstream. During the study, 770 lock cycles were completed, with 58,982 fish being observed in 234 cycles, mostly moving upstream. Eels (*Anguilla anguilla*) were the most numerous, being followed by the mugilids (*Chelon ramada* and *Mugil cephalus*). The number of fish using the lock was low from November to April, increasing from May to July, and was related to tide height, discharge at the powerplant tailrace, moon phase and time of day. Sea lamprey (*Petromyzon marinus*) and shads (*Alosa* spp.) were caught downstream of the dam but were never found using the lock. While more research is needed to assess the fate of the eels moving upstream, management of the anadromous taxa in the Douro River must rely on the last 20 km of the river.

**Keywords:** fish lock; hydropower; migration; Iberia; Douro River

## 1. Introduction

Fish locks are used in impassable weirs and dams to improve fish upstream displacements. The idea of a fish lock was originally suggested in 1890 [1] and the first one was constructed in the USA in the 1930s [2]. Following this initial effort, J. Borland developed the design of a fish lock, with the first pass of this type built at the Leixlip Dam (River Liffey, Ireland) in 1949. Soon after, several Borland fish locks were built in Scotland and Ireland in the 1950s and 1960s that are presently still in use. Around 80 fish locks were installed throughout the world, the majority in Europe (Austria, Germany, Ireland, Portugal, Russia, Scotland and Spain), but also in America and Australia [2,3]. In short, a fish lock consists of a downstream lower chamber and an upper chamber that are connected by either a sloping canal or a vertical well, with sluice gates fitted to each end to control the lock operation. The operating principle of a fish lock is very similar to that of a navigation lock [4]; fish are attracted into the downstream chamber, and then pass through the lock in the same way as a boat. Each lock operating cycle begins with an attraction/fishing phase when the downstream gate is open, and the upstream gate controls the flow into and through the lock. During this phase, which can last variable periods of time, fish are expected to enter the downstream chamber in response to the flow generated. After that, during the filling and exit phase, the downstream gate closes and the lock fills up. The fish are then expected to follow the free surface of the water in the canal, rising and reaching the upstream chamber when the lock is full. The fish are encouraged to pass into the upstream water body following the attraction flow generated by the partially opened upstream gate. Finally, the upstream gate is closed and the lock is emptied. When emptying is almost complete and the head on the downstream sluice is low enough, the downstream sluice

is re-opened. The duration of each lock cycle can vary from less than one hour to four hours [2,5].

Guidance on design criteria and recommendations for operation for fish locks is sparse, and as a result existing structures are somewhat unique, likely influencing their use by fish [2]. The efficiency of fish locks, as of other fishway types, is dependent on the behavior of the fish [5]. They must remain in the downstream chamber during the attraction phase, follow the rising water level during the filling stage and then leave the lock before it empties. Moreover, a lock operates discontinuously, limiting the total numbers of fish using these devices, and the attraction flow generated by the opening of the upstream gate is usually small, which can reduce the attractability for fish to enter, so they stay in the downstream chamber.

In Europe, the usability of these locks by fish has been assessed with variable outcomes. In France, most fish locks were considered ineffective for fish use, notably anadromous allis shad (*Aloso alosa*), being more recently replaced by fish lifts [5]. In contrast, in Scotland and Ireland, fish locks are still used today, with success for salmonids in several dams and weirs [2]. Elsewhere, available results revealed also rather variable performance. In the USA, the first locks constructed at dams on the Columbia River were abandoned in favor of pool fish passes [5]. However, the lock installed in the St. Stephen power plant (Santee River, South Carolina) in 1985 is used by fish like the American shad (*Alosa sapidissima*) and blueback herring (*A. aestivalis*) during their annual upstream migration, with more than 500,000 fish using the lock in some years (https://www.dnr.sc.gov/fish/fishlift/stats.html (accessed on 30 December 2022)). In South America, fish locks are also present. In the Salto Grande dam (Uruguay river, Argentina and Uruguay), two fish locks operate, but results showed that they cannot dependably pass large numbers of migratory species [6]. There are eight fish locks in Australia, with six of these in tropical/ sub-tropical coastal rivers of the northeast [7]. The fish lock installed in a weir in the tropical Fitzroy River was used by fish, with a maximum rate of 3317 fish per day [7], and the authors considered fish locks to have considerable potential for tropical river systems with low minimum flows and low biomass. In the Iberian Peninsula, several dams were equipped with fish locks in the 1970s and 1980s, most notably the first five cascading dams built in the Douro River from 1971 to 1985. In Europe, most available references concerning the fish utilization of fish locks are old and targeted mostly salmonid rivers [3]. Except for smaller headwater streams in the northern part of the Iberian Peninsula, where brown trout (*Salmo trutta*) may dominate fish assemblages, Iberian freshwater fish communities are characterized by native Cypriniformes and amphidromous taxa, but the knowledge about the use of fish locks by these taxa is less investigated.

In this study, the fish use of the Borland fish lock installed in the most downstream dam in the Douro River was investigated over nine months. Moreover, a creel survey was conducted encompassing the same period to characterize fish catches by the professional fisherman community fishing downstream of the dam. More specifically, this study aimed to: (i) assess the temporal variation in the fish use of the Borland fish lock, (ii) identify environmental triggers of the lock use by different species, and (iii) assess selectivity in the use of the lock by different species by contrasting the frequencies in the lock and in the catches made downstream.

## 2. Materials and Methods

### 2.1. Study Area

The studied fish lock was installed in the Crestuma-Lever dam during its construction. This is the farthest downstream dam in the Douro River, the third longest Iberian River, and was the last built in the main river, beginning its operation in 1985 (Table 1, Figure 1). It is installed 20 km upstream of the river mouth, at the Atlantic Ocean.

**Table 1.** Main characteristics of the Crestuma-Lever dam/reservoir and the Douro River. Data sources: dams in Portugal (https://cnpgb.apambiente.pt/gr_barragens/gbportugal/Crestuma.htm (accessed on 30 January 2022)) and Douro River basin management plan (https://www.apambiente. pt/node/1598 (accessed on 30 January 2022)).

| Dam and Reservoir | |
| --- | --- |
| Installed capacity | 117 MW (three Kaplan turbines) |
| Maximum discharged volume at each turbine | 440 m$^3$/s |
| Height of the crest | 25.50 m |
| Average flow at the dam | 450 m$^3$/s |
| Inundated area at pool level | 12.98 km$^2$ |
| Total volume | 22.5 hm$^3$ |
| Douro river | |
| Total length | 897 km |
| Catchment area | 97,600 km$^2$ |
| Average total discharge | 11,349 hm$^3$ |

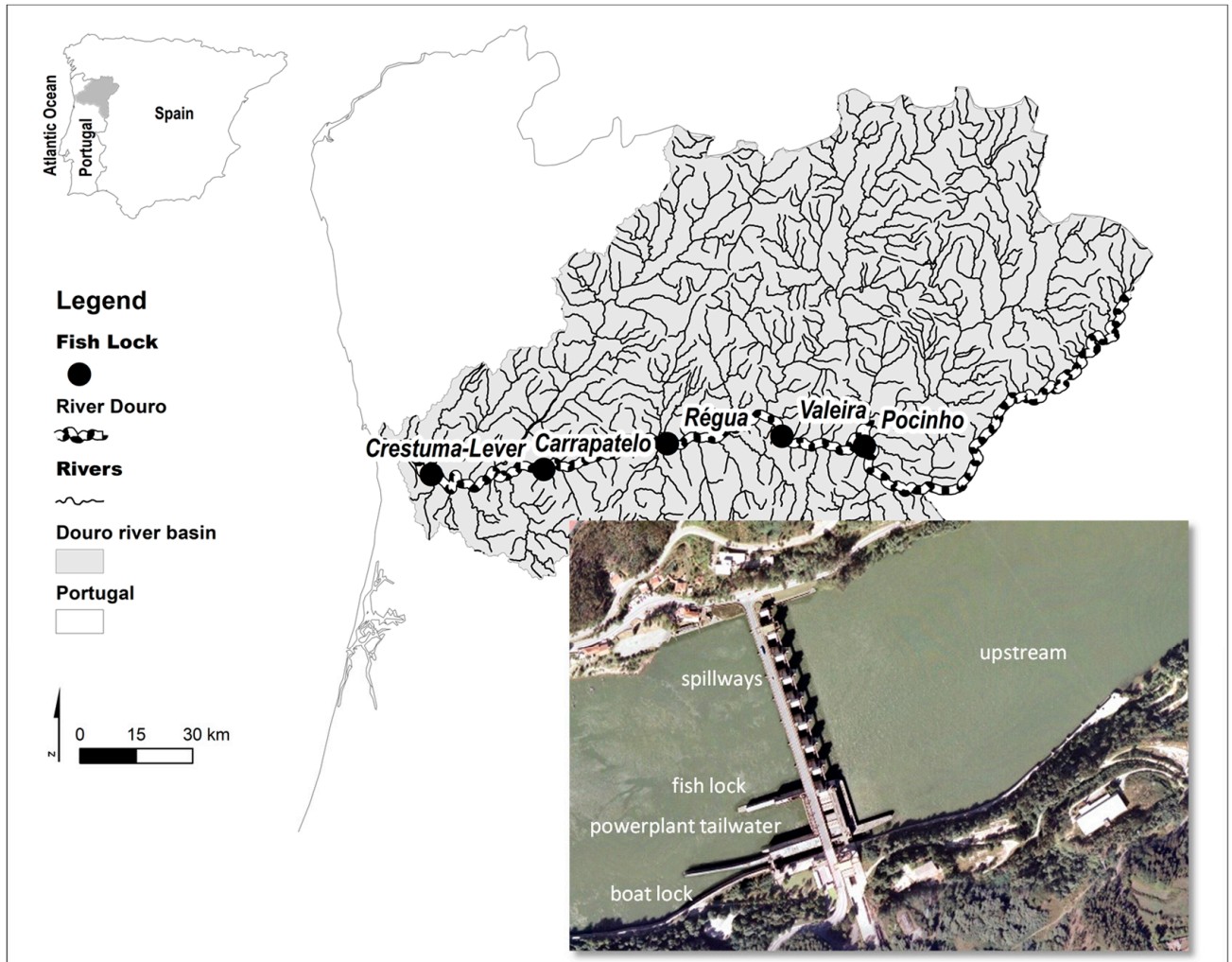

**Figure 1.** The last 325 km of the Douro River showing the existing dams in Portugal. Insert: Crestuma-Lever dam, depicting the spillways, powerplant tailwater and the fish and boat locks (Image Source: Municípia S.A.).

The four upstream dams (Carrapatelo, operating since 1971; Régua, operating since 1973; Valeira, operating since 1976; and Pocinho, operating since 1982) are also equipped

with Borland fish locks. In addition to the fish locks, the dams have also boat locks to allow navigation in the river.

The river stretch below the Crestuma-Lever dam forms a mesotidal estuary, with an average spring tidal range of 2.8 m near the dam [8]. Salinity varies mostly in relation to the amount of river flow, reaching higher values during summer lower flows (18 psu near the dam at 39 m$^3$/s) and lower values during winter higher flows (<0.5 psu at 270 m$^3$/s) [9].

As in other Iberian river basins, the Douro River fish fauna is characterized by the presence of native Cyprinidae and Leuciscidae, but biotic integrity has been decreasing in recent decades in response to flow regulation and the successful introduction of several exotic species, including widespread species such as the largemouth bass (*Micropterus salmoides*), pumpkin-seed sunfish (*Lepomis gibbosus*), bleak (*Alburnus alburnus*) and pikeperch (*Sander lucioperca*) in the reservoirs, and the bleak and Iberian gudgeon (*Gobio lozanoi*) in the rivers [10–13]. Moreover, recent introductions are spreading in the basin, namely the wels catfish, *Silurus glanis,* and the North American catfish, *Ameiurus melas* [12]. The native cyprinids include potamodromous species, notably the Iberian barbel (*Luciobarbus bocagei*) and the Northern straight-mouth nase (*Pseudochondrostoma duriense*).

Salmonids are only present in headwaters, although some Atlantic salmon (*Salmo salar*) are still caught below Crestuma-Lever dam every year [14]. In the Douro estuary, that presently stretches to the Crestuma-Lever dam, several catadromous and anadromous species also occur, including the allis shad, the sea lamprey (*Petromyzon marinus*), the European eel (*Anguilla anguilla*), the mullets—thinlip grey mullet (*Chelon ramada*) and flathead grey mullet (*Mugil cephalus*)—and the European flounder (*Platichthys flesus*), as well as some primary marine species such as the European sea bass (*Dicentrarchus labrax*) [13,15]. Migratory species that are presently mostly restricted to the river stretch below Crestuma-Lever dam had much larger distribution areas in historic times, reaching up to 260 km upstream of the Douro River mouth [10,16].

### 2.2. Fish Lock Characteristics and Operation

The Borland fish lock is installed at the Crestuma-Lever dam between the turbines' tailwater and the spillways (Figure 1). It is constituted of a lower chamber (12.0 m long × 3.5 m width × 8 m height), a vertical well (13.0 m height × 3.5 m diameter) and an upper chamber that connects with the reservoir through a canal (55.0 m long × 1.5 m width).

The entrance to the lock is made through two vertical inlets (0.75 m width) facing the spillways, each with a different lower level to ensure operation at variable downstream levels. To the side of each entrance there are four small outlets that can discharge part (30%) of the lock attraction flow.

Water levels and flow in the lock are controlled through sluice gates installed at the upstream canal and at the two inlets. In 2015 and 2016, a series of interventions were carried out by the company that operates the dam to automate the operation of the fish lock, allowing the continuous succession of cycles, the stopping of cycles when the downstream levels drop below a determined value and the continuous record of each cycle stage and other operational details.

The duration of each lock cycle and other operational characteristics of the Crestuma-Lever fish lock are presented in Table 2.

Fish were observed during each passing phase at a viewing window that is installed in the upper lock chamber (Figure 2). A white board with a 20 cm rectangular black frame was placed opposite to the viewing window to increase contrast and to aid in the approximate estimation of fish length.

**Table 2.** Operational details for the fish lock of the Crestuma-Lever dam (Data provided by the dam owner, EDP).

| Phase Duration | |
|---|---|
| Attraction/fishing phase | 60 min |
| Filling stage | 20–25 min |
| Passing stage | 45 min |
| Emptying phase | 25–30 min |
| **Other characteristics** | |
| Flow during the attraction phase | 0.5–1.0 m³/s |
| Velocity at the entrance during the attraction phase (variable according to the downstream water level and the position of the upstream sluice gate) | 1.2–2.6 m/s⁻¹ |
| Velocity inside the lock during fish passing | 0.3 m/s |

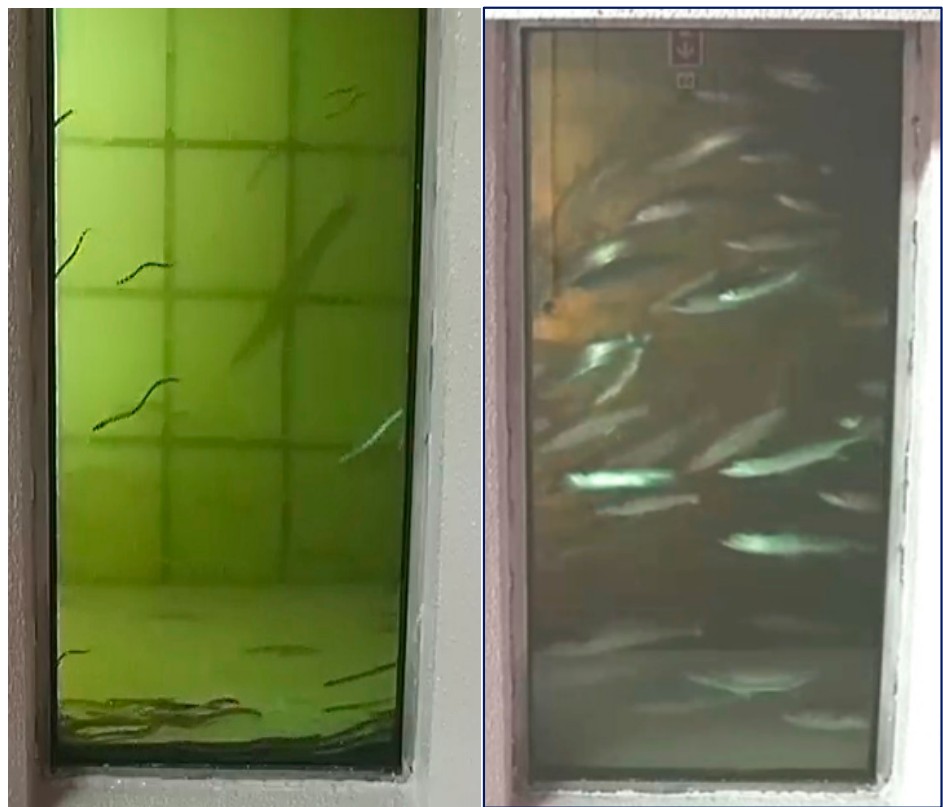

**Figure 2.** Viewing window in the upper lock chamber with several European eels (**left**) and mullets (**right**) passing upstream.

### 2.3. Data Acquisition during Lock Cycles

The monitoring of the fish lock began in November 2016 and ended in July 2017. A video system was installed at the viewing window to continuously record images during the passing stage (HIKVISION™ camera, model DS-2CE16C2T-VFIR3 with infrared and varifocal lens, and a digital video recorder HIKVISION™). Video data were stored in 2TB hard drives and were accessed and downloaded through an Internet server.

Video files from the passing stage of each lock cycle were visually analysed with the software PotPlayer™ (https://potplayer.daum.net/). Fish using the lock were identified to species whenever possible and counted during each cycle. Some smaller specimens of Cyprinidae and Leuciscidae could not be identified to species and were grouped as non-identified Cypriniformes. Mugilids could also not be identified to species and were grouped. The movement direction was registered for each individual. Some specimens,

particularly the larger ones, were approximately measured by using the background black reticulate and the width of the viewing window (0.595 cm).

In addition to the fish using the lock, several explanatory variables were obtained, including the tide height at the beginning of each cycle, the moon phase, the circadian period (night, from 08:00 p.m. to 07:59 a.m., or day, from 08:00 a.m. to 07:59 p.m.) and the discharged flow at the powerplant tailwater during the previous 24 h. Two different lock cycles were assessed, either including or not the flow generated by the lateral outlets.

### 2.4. Fish Captures by Professional Fishermen Downstream the Dam

A creel survey was conducted with the professional fisherman fishing downstream of the dam between August 2016 and July 2017. Fishermen were interviewed weekly, usually during Friday night or Saturday, and were asked about the catch made during the previous week, including the number of days fished (usually during the night), species caught and their numbers. The fishermen use gillnets and trammel nets to target mostly allis shad and sea lamprey, which have an open fishing season from 1 January to the 30 April, but also catch other species that are vulnerable to the nets used (floating gillnets with 70 mm knot to knot and trammel nets with an inner panel of 100 mm knot to knot). Other fishing methods are used to target additional species, particularly the European eel, including longlines and several small-scale traditional fishing methods such as the *rapeta* and *remolhão*.

The total number of each fish taxa caught monthly was divided by the number of fishing days reported by the interviewed fishermen to derive an abundance indicator from the catch. The typical fishing day usually encompasses fishing throughout the night, including repeatedly setting and lifting the nets. The exact duration can vary according to the fishing success and the discharged flow at the powerplant tailwater.

### 2.5. Data Analysis

The number of fish specimens per lock cycle was used to describe the lock functioning and was log-transformed with log (x + 1) as it showed marked right-skewness.

One-way analysis of variance (ANOVA) was initially used to assess the effects of the following dependent variables in the fish use of the lock (response variable): time of day (day or night), month, tide height (two classes, for tides above or below 2 m in height), moon phase (new moon, first quarter, full moon, and last quarter), discharged flow at the powerplant (two classes, above and below 200 $m^3/s^{-1}$) and cycle type (cycle 1, without flow through the lateral outlets, and cycle 2, with flow through the outlets). The fit of each ANOVA was assessed with adjusted R-squared.

Further, to investigate the influence of relevant variables on the fish use of the lock, general linear models (GLM) were adjusted to explanatory variables accounting for at least 5% of the variance in the number of fish visualized in the lock, potentially including both continuous (tide height and discharged flow) and categorical (lunar phase, cycle type, month and time of day) variables. Partial $\eta^2$, which measures the proportion of variance explained by a given variable of the total variance remaining after accounting for the variance explained by other variables in the model, was used to assess effect size.

All statistical analyses were run using STATISTICA [17].

## 3. Results

### 3.1. Fish Lock

The number of cycles made varied considerably among the studied period, reaching a maximum of 240 in December (Table 3). In February, no cycle was made, as the fish lock was blocked upstream due to the debris transported during an extended flood period. In addition, from February to May, malfunctioning of the fish lock limited the number of cycles made, whereas the river level downstream, as influenced by the tide height, restricted the operation of the lock on other occasions. On a single day, a maximum of nine cycles were performed.

**Table 3.** Number of fish counted at the Crestuma-Lever fish lock between November 2016 and July 2017 (U—upstream, D—downstream).

| Month | Total Number of Cycles | Number of Cycles with Fish | Mugilidae | | European Eel | | Iberian Barbel | | Pumpkin Seed Sunfish | | Largemouth Bass | | Bleak | | European Flounder | | Unidentified Cypriniformes | |
|---|---|---|---|---|---|---|---|---|---|---|---|---|---|---|---|---|---|---|
| | | | U | D | U | D | U | D | U | D | U | D | U | D | U | D | U | D |
| November 2016 | 72 | 12 | 0 | 0 | 1 | 1 | 0 | 0 | 11 | 1 | 0 | 0 | 0 | 0 | 0 | 0 | 0 | 0 |
| December 2016 | 240 | 44 | 6 | 1 | 160 | 3 | 0 | 0 | 0 | 3 | 0 | 0 | 17 | 0 | 0 | 0 | 6 | 5 |
| January 2017 | 116 | 10 | 0 | 0 | 158 | 0 | 0 | 0 | 0 | 0 | 0 | 0 | 0 | 0 | 0 | 0 | 0 | 0 |
| March 2017 | 85 | 1 | 79 | 0 | 11 | 1 | 0 | 0 | 0 | 0 | 0 | 0 | 0 | 0 | 0 | 0 | 0 | 0 |
| April 2017 | 82 | 1 | 0 | 0 | 3 | 0 | 0 | 0 | 0 | 0 | 0 | 0 | 0 | 0 | 0 | 0 | 0 | 0 |
| May 2017 | 16 | 11 | 3264 | 0 | 8 | 0 | 1263 | 0 | 0 | 0 | 0 | 0 | 0 | 0 | 0 | 0 | 0 | 0 |
| June 2017 | 13 | 13 | 6491 | 0 | 177 | 0 | 0 | 0 | 0 | 0 | 0 | 0 | 17 | 0 | 0 | 0 | 0 | 0 |
| July 2017 | 146 | 142 | 2719 | 2 | 44,104 | 1 | 9 | 0 | 159 | 0 | 9 | 0 | 221 | 7 | 3 | 0 | 55 | 6 |
| TOTAL | 770 | 234 | 12,559 | 3 | 44,622 | 6 | 1272 | 0 | 170 | 4 | 9 | 0 | 255 | 7 | 3 | 0 | 61 | 11 |

During the study period, 770 cycles were completed, with fish being observed in 234 (30.4%). The frequency of fish occurrence in the cycles made varied during the studied months. It was low in March and April, when less than 1.3% of the cycles had the presence of fish, clearly increasing from May to July.

In total, 58982 fish individuals were counted moving through the lock (Table 3). European eel was the most numerous species using the fish lock (75.7% of all fish counted), being followed by mugilids (21.3%). Other species found included Iberian barbel, bleak, European flounder, pumpkinseed sunfish, largemouth bass and unidentified Cypriniformes.

The vast majority of the fish passages were made upstream, but some downstream displacements were also noted (0.05% of the total number of fish counted). The number of fish using the lock was low from November to April, increasing from May to July in response to the upstream displacements of European eel and mugilids (Figure 3). The majority of the Iberian barbel using the lock was recorded in just two cycles, during the day on 12 May, when 1244 large-sized specimens (>400 mm in total length, TL) moved upstream.

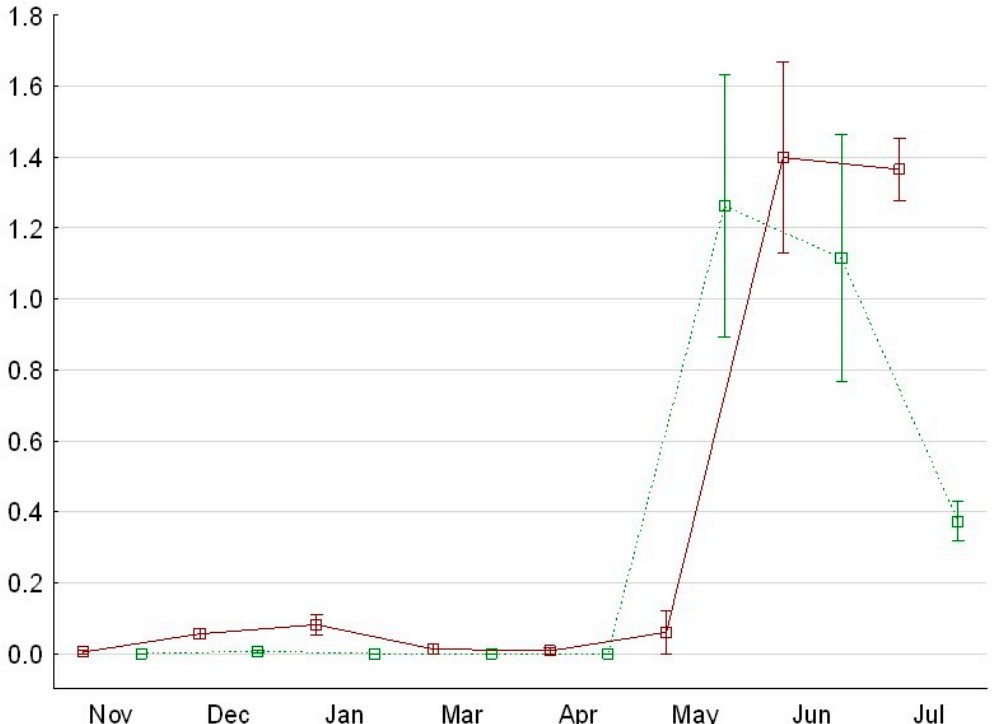

**Figure 3.** Number of individuals of the two main taxa using the fish lock (mugilids and European eel) per cycle (mean of log10(number+1) ±SE) in the investigated months (Continuous red line—European eel, interrupted green line—mugilids).

Several fish dimensions were observed in the Crestuma-Lever fish lock, from small specimens of leuciscids (<50 mm in TL) to Iberian barbel and mugilids above 600 mm in TL. A Eurasian otter (*Lutra lutra*) was recorded once inside the fish lock, entering from upstream. European eels passing from November to January were small sized (<150 mm in TL), including elvers, whereas individuals moving later included larger specimens.

Except for cycle type, all the explanatory variables considered were related to differences in the number of European eels and mugilids using the fish lock (Table 4).

**Table 4.** Main results of the ANOVA for the number of individuals per cycle (ind.cyle$^{-1}$) according to explanatory variables (NS—non-significant).

| European Eel | | | |
|---|---|---|---|
| | F | *p* | Adjusted R$^2$ |
| Day/night | 4.84 | <0.05 | 0.01 |
| Moon phase | 40.41 | <0.01 | 0.1 |
| Cycle type | 0.06 | NS | - |
| Month | 125.14 | <0.01 | 0.53 |
| Tide height | 8.66 | <0.01 | 0.02 |
| Discharged flow | 295.9 | <0.01 | 0.27 |
| Mugilids | | | |
| | F | *p* | Adjusted R$^2$ |
| Day/night | 15.74 | <0.01 | 0.02 |
| Moon phase | 6.74 | <0.01 | 0.01 |
| Cycle type | 0.21 | NS | - |
| Month | 49.2 | <0.01 | 0.31 |
| Tide height | 8.07 | <0.01 | 0.02 |
| Discharged flow | 68.85 | <0.01 | 0.09 |

European eels used the fish lock during all months studied, but were particularly abundant in July, when 44104 individuals (302 ind.cycle$^{-1}$) were counted (Figure 3). Duncan post hoc tests showed significant differences ($p < 0.05$) in the number of individuals per cycle between June and July and all the other months. Eels used the lock more at night (ind.cycle$^{-1}$, night—0.37, day—0.26) when there was a full moon (ind.cycle$^{-1}$, new moon—0.24, first quarter—0.01, full moon—0.81 and last quarter—0.19, significant differences between all moon phases, except new moon and last quarter according to Duncan tests), after lower amounts of flow at the powerplant tailwater (ind.cycle$^{-1}$, lower flows—0.86, higher flows—0.06) and during higher tides (ind.cycle$^{-1}$, lower tides—0.14, higher tides—0.37).

Mugilids utilized the fish lock mostly from May to July, when a total of 12474 individuals moved upstream (Duncan tests showed significant differences in the number of individuals per cycle between May and June and all the other months, whereas July was distinct from all the other months). As for European eel, mugilids used the lock more after lower amounts of flow at the powerplant tailwater (ind.cycle$^{-1}$, lower flow—0.26, higher flow—0.03), during higher tides (ind.cycle$^{-1}$, lower tides—0.01, higher tides—0.14) and during the full (but also new) moon (ind.cycle$^{-1}$, new moon—0.16, first quarter—0.01, full moon—0.15 and last quarter—0.05, significant differences between all moon phases, except new moon and full moon, and first and last quarter according to Duncan tests). However, in contrast to European eels, mugilids used the lock more heavily during the day (ind.cycle$^{-1}$, night—0.05, day—0.18).

In general, the explanatory variables used in ANOVA provided a better fit for the European eel than for the mugilids, with the sampling month accounting for larger portions of the variance in the number of individuals using the lock for both eel and mugilids.

The results of the GLM fitted to eel and mugilids were in line with the ANOVA output (Table 5), highlighting the importance of the sampling month for the results.

**Table 5.** Main results of the general linear models relating the ind.cyle$^{-1}$ according to the explanatory variables accounting at least for 5% of the variance. Significant factors ($p < 0.05$) are denoted in bold. Adjusted r-squared were 0.54 and 0.31, respectively, for eel and mugilids.

| European Eel | Month | Discharged Flow | Moon Phase |
|---|---|---|---|
| F | **57.4** | 0.54 | **15.75** |
| Partial $\eta^2$ | 0.35 | <0.01 | 0.06 |
| Mugilids | Month | Discharged flow | Moon phase |
| F | **36.2** | 0.26 | |
| Partial $\eta^2$ | 0.25 | <0.01 | |

*3.2. Creel Survey*

A total of 377 fishing days were reported by the professional fisherman. The number of fishing days increased from December before reaching a peak in April (Figure 4). More than 60% of the fishing days reported occurred in March and April. As expected, the total catch increased with the fishing effort (Pearson R between the monthly catch and the number of fishing days/month = 0.87, $p < 0.01$, Figure 4).

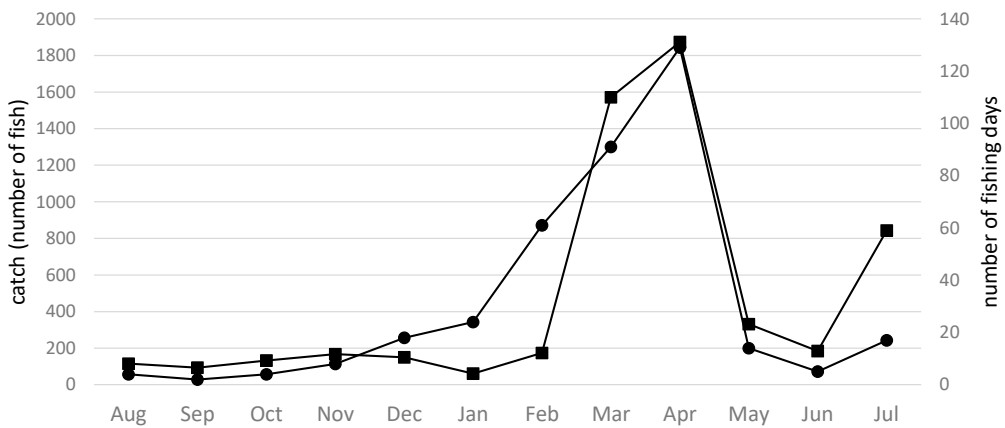

**Figure 4.** Number of fishing days (circle marker) and captures (square marker) reported by the fishermen between August 2016 and July 2017.

During the study period, 14 species were caught by the fisherman. The most frequent catches included mugilids, allis shad, sea lamprey, European eel and Iberian barbel. Other species caught included largemouth bass, pikeperch, twaite shad (*Alosa fallax*), Eurasian carp (*Cyprinus carpio*), bleak, sea bass, European flounder and a single specimen of Atlantic salmon. The anadromous shads and sea lamprey were mostly collected between February and April, representing more than 50% of all catches in February.

Excluding the European eel, as it was reported in weight, mugilids were the numerically dominant catch downstream of Crestuma-Lever dam, being caught during all months, except December to February (Figure 5, Table 6); catches of this species reached 20 ind.fishing day$^{-1}$. Sea lamprey was caught from December to May, with higher catches in March. Shad catches were also clearly seasonal, being caught between December and June and reaching a peak in March. During the studied year, allis shad catches were higher than sea lamprey catches, which according to the fishermen is sometimes reversed. The Iberian barbel was caught over nine months, particularly during August, October and November.

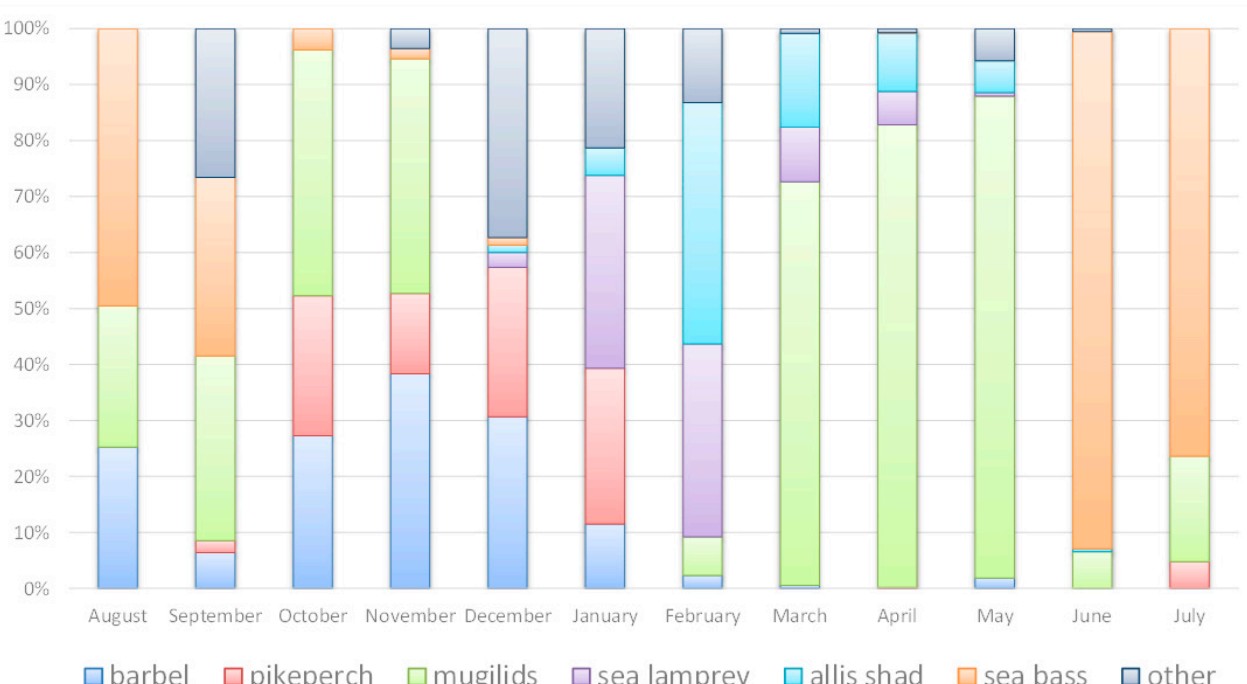

**Figure 5.** Proportion of the monthly total catch (number of individuals) represented by the different species caught downstream of the Crestuma-Lever dam from August 2016 to July 2017 (European eel is not included as its catches were reported in weight).

**Table 6.** Average catch (ind.fishing day$^{-1}$) by fisherman downstream of the Crestuma-Lever dam from August 2016 to July 2017 (* including bleak, largemouth bass, Eurasian carp, Atlantic salmon and two-banded seabream, *Diplodus vulgaris*).

| | Iberian Barbel | Pikeperch | Mugilids | Sea Lamprey | Allis Shad | Twaite Shad | European Sea Bass | European Flounder | Others * |
|---|---|---|---|---|---|---|---|---|---|
| Aug | 7.3 | 0.0 | 7.3 | 0.0 | 0.0 | 0.0 | 14.3 | 0.0 | 0.0 |
| Sep | 3.0 | 1.0 | 15.5 | 0.0 | 0.0 | 0.0 | 15.0 | 1.5 | 11.0 |
| Oct | 9.0 | 8.3 | 14.5 | 0.0 | 0.0 | 0.0 | 1.3 | 0.0 | 0.0 |
| Nov | 8.0 | 3.0 | 8.8 | 0.0 | 0.0 | 0.0 | 0.4 | 0.0 | 0.8 |
| Dec | 2.6 | 2.2 | 0.0 | 0.2 | 0.1 | 0.1 | 0.1 | 1.4 | 1.7 |
| Jan | 0.3 | 0.7 | 0.0 | 0.9 | 0.1 | 0.0 | 0.0 | 0.3 | 0.2 |
| Feb | 0.1 | 0.0 | 0.2 | 1.0 | 1.2 | 0.0 | 0.0 | 0.4 | 0.0 |
| Mar | 0.1 | 0.0 | 12.5 | 1.7 | 2.9 | 0.0 | 0.0 | 0.2 | 0.0 |
| Apr | 0.0 | 0.0 | 12.0 | 0.9 | 1.5 | 0.1 | 0.0 | 0.0 | 0.0 |
| May | 0.4 | 0.0 | 20.4 | 0.1 | 1.4 | 1.4 | 0.0 | 0.0 | 0.0 |
| Jun | 0.0 | 0.0 | 2.4 | 0.0 | 0.2 | 0.0 | 34.0 | 0.0 | 0.4 |
| Jul | 0.0 | 2.4 | 9.4 | 0.0 | 0.0 | 0.0 | 37.9 | 0.0 | 0.0 |

European eels were caught in December (4.0 kg), January (3.0 kg), February (1.0 kg) and May (0.7 kg). Notwithstanding, the reported values should be considered with caution and are likely underestimated, as illegal fishing for elvers occurs in the study area.

### 3.3. Selectivity in the Use of the Fish Lock

When comparing the fish caught downstream of the Crestuma-Lever dam and the fish visualized in the lock, the most striking difference is related to the anadromous shad and sea lamprey, which despite being captured by fisherman between December and May were never found in the fish lock during the same months. In contrast, other species frequent in the catches, notably the mullets, European eel and Iberian barbel, used the lock.

## 4. Discussion

### 4.1. Temporal Use of the Fish Lock

Fish utilized the Crestuma-Lever lock irregularly throughout the examined period, as previously reported in a preliminary study performed in the same lock from the end of October 1994 to the middle of March 1995 [18]. The use was extremely infrequent from November to April, clearly increasing in May through July, when most lock cycles were used by fish to pass upstream. In our study, 98% of all the fish observed using the lock passed upstream from May to July, whereas [18] reported that almost no fish used the lock from December to the beginning of March. Accordingly, sampling month was the main factor explaining the variation in the number of European eels and mugilids using the Crestuma-Lever fish lock.

Similarly, higher fish usage at other fish passes throughout the temperate region has been reported for comparable periods. In another Iberian hydropower dam (Touvedo in the Lima River, Northern Portugal), 70% of the 1194 fish using a fish lift during a year, including leuciscids, cyprinids, salmonids and European eel, were observed between May and July [19], while a similar movement pattern was reported for the fish lift in the Tuilieres hydropower dam (Dordogne River, France) [20]. In a series of studies conducted in the vertical-slot fish pass installed in the Ponte-Coimbra weir, the first insurmountable obstacle to migration at the Mondego River (Center of Portugal) located approximately 45 km upstream from the river mouth, most fish passing were recorded during spring and summer [21,22]. Further north in Europe, the lock installed in the Ardnacrusha hydropower dam (Ireland) was little used by fish, including salmonids and European eel, during the first quarter of the year [3,23].

The Crestuma-Lever fish lock, as with other fish passes, is expected to be used more frequently as fish movement increases in rivers and estuaries, usually within migrations, i.e., synchronized movements by populations or population components driven by the transitory availability and changing location of key resources [24]. Although other dispersal movements are made by individual fish, most movements of large numbers of fish are migratory displacements, and the majority of the fish using the Crestuma-Lever fish lock were the catadromous European eel, mugilids, and the potamodromous Iberian barbel.

Overall, the increase in fish migratory movements in the temperate region has been linked to increases in water temperature, usually during the spring [19,22,25–27], and water warming likely triggered the increasing use of the fish lock installed at Crestuma-Lever dam throughout the study period.

The European eel used the lock particularly during July, although it was observed in every month examined in this study. Similarly, in the Touvedo fish lift, yellow eels were transported upstream mostly during July [19], although in much lower numbers than at Crestuma-Lever (1.7 ind.cycle$^{-1}$ at Touvedo and 302 ind.cycle$^{-1}$ at Crestuma-Lever). In the fish lock installed in the Ardnacrusha hydropower dam, the major European eel upstream displacements extended from June to August [3]. All these results are concordant with the reported reduced activity at colder temperatures and the increased upstream movement of juvenile European eels with increasing temperature [25,26,28,29].

The mullets used the Crestuma-Lever lock with particular intensity during June (499 ind.cycle$^{-1}$). In the vertical-slot fish pass installed in the Ponte-Coimbra weir, the thinlip mullet (*Chelon ramada*), the main species using the pass, moved upstream during June (dryer years) and July (wetter years) [22]. Where they occur, mugilids have been found to extensively use different types of fish passes and to start migrating upstream to freshwater feeding habitats during the spring, peaking in the summer [30–34]. As for European eel, temperature has been identified as a trigger for the migrations of mugilids, with upstream movements increasing with temperatures from 15 °C to 20 °C [22].

In a single passage, 1180 large sized Iberian barbel moved upstream through the Crestuma-Lever lock in May. Our results agree with other studies, showing that native barbel migrate upstream of Iberian rivers during spring, with the mass migration of adults to access spawning habitats reported for mid-May elsewhere [35]. It is not known at this

time whether the Iberian barbel using the lock remain in the Douro estuary all year or if they migrate to the estuary from tributary streams (e.g., River Sousa).

### 4.2. Factors Influencing Lock Utilization by Fish

In addition to the sampling month, as it relates with changes in water temperature, other explanatory variables were linked to the use of the fish lock by the two most common species transposing the Crestuma-Lever dam, despite presenting a smaller influence on the results.

Discharge was the second most important explanatory variable linked with the use of the lock by European eel and mugilids, although its effect lack significance when taking into consideration the effect of sampling month, as a decrease in discharged flow from winter to summer closely follows the increased fish use of the Borland fish lock during the same period. Similar to our findings, the influence of discharge has been identified as a trigger for fish migration in rivers and estuaries, e.g., [26,36–38], with studies reporting larger upstream movements with larger flows, e.g., [39,40], but also higher movements with lower flows [41].

Some studies found that upstream migration of elvers and yellow eels (*Anguilla* spp.) coincided with increasing water flow [25,26,42], although movement could temporarily stop at higher flows [43]. Catadromous migrations of mugilids to freshwater feeding habitats have been linked to summer reduced river discharges [22]. Most of the river flow downstream of the Crestuma-Lever dam is generated at the powerplant tailwater, as the spillways are used to discharge excess river flow in just a few days per year and only in wetter years, and in the present study European eels and mullets used the lock almost exclusively (98% of all the fish) when the powerplant was operating well below its maximum capacity. In the Ponte-Coimbra weir, the thinlip grey mullet upstream movements through the fish pass were restricted by dam discharges higher than $160 \, \text{m}^3/\text{s}^{-1}$ [22], while shads and sea lamprey were restricted by much lower discharges [21,44].

Nonetheless, the attractability of fish passes is linked to the specific flow patterns generated in the vicinity of the entrances, e.g., [21,22,44,45]. It is possible that the flow pattern generated near the powerplant tailrace with higher discharges had reduced the attractivity of the Crestuma-Lever fish lock. Although the entrance to the lock is directed towards the spillways and not to the powerplant tailwater, the high discharge generated when the hydropower plant is in full operation could produce confounding turbulence near the lock entrance. Moreover, by creating a main flow axis not coincident with the lock entrance, higher flows of the powerplant tailwater could have lowered the fish pass attractiveness by masking the attraction flow [5]. However, the flow pattern near the lock entrance, as in other areas downstream of the Crestuma-Lever dam, is unknown, precluding a better understanding of fish behaviour in the vicinity of the lock entrance.

Tide height was also related to the number of fish using the lock, with more European eel and mugilids passing the dam during higher tides. The influence of tide height on the movement of fish in estuaries has been described various times. For example, in the Mira River estuary (Portugal), movement direction of adult thinlip grey mullet was strongly dependent upon the tidal state, with upstream movements being made during the flood [46], while the European flounder moved upstream the Bann River (Ireland) during rising and high tides [47]. Moreover, upstream movement of young-of-the-year from several catadromous fish species, including European eel, European flounder and thinlip grey mullet use selective tidal-stream transport (STST) during the tidal estuary crossing, with individuals being transported horizontally by entering the water column during the flood and descending to the bottom during the ebb [48].

Time of day and moon phase, particularly for the European eel, also influenced how fish used the Crestuma-Lever lock. Yellow eels used the Touvedo fish lift more often during the night with new moons [29] and several studies have reported night peak migrations of *Anguilla* spp. with low moon illumination [25,26,49,50]. Concordantly, in the Crestuma-Lever, European eel used the lock more at night but, opposite to these findings, also during

periods of higher moon luminosity (i.e., full moon). It is possible that the higher tide height linked with the new moon had overridden the effect of the moon phase/luminosity on the passing of European eels in the Crestuma-Lever lock. The higher use of the lock by mullets during the full and new moon was also likely linked to the influence of tide height, as their use of the lock was mostly diurnal. Elsewhere, mullet movements have been reported more often or even exclusively during daylight hours, e.g., [22,31,51], with this being linked to their feeding activity [51–53].

The two lock cycles tested did not yield different results in the number of European eel and mullets using the lock. While the use of the lateral outlets reduced the flow in the Crestuma-Lever lock entrance, the reduction is small and the hydraulic conditions existing were largely similar for the two cycle types examined.

*4.3. Selectivity of the Borland Fish Lock*

The low utilization of the lock during some months outside the migration period of the main studied fish species (European eel and mullets) was likely influenced by the absence of migratory stimulus to move and pass the obstacle, but the lock was never used by the anadromous species targeted by professional fishermen during their migration period, although according to the creel survey these species were present below the dam at least between March and May, when 167 lock cycles were completed.

The study conducted by [18] also failed to find any specimens of shads and sea lamprey in the Crestuma-Lever fish lock. Sea lamprey and shads use other fish passes in Iberia. During a four-year study, nearly 50,000 lampreys successfully moved upstream through the Ponte-Coimbra fish pass [44] and more than 26,000 allis and twaite shads did the same from 2013 to 2017 [21]. However, the Mondego is considered one of the last strongholds for these species in Portugal [54], whereas the Portuguese red list of endangered fish species considers the shad population to be inviable in the Douro River due to the cascade of dams [55].

It is clear that the anadromous species still occurring in the Douro (shads, sea lamprey and Atlantic salmon) have been decreasing in numbers following the construction of the hydropower dams in Portugal and further upstream in Spain. The historic distribution areas of these species have been drastically reduced and remaining populations have to persist by using only the last 20 km of the river. Although captured by the fisherman, the anadromous fish catch was rather low, which justifies the complementary nature of the fishing activity below the Crestuma-Lever dam, as most fisherman have additional jobs. Nevertheless, fish from these species might have difficulties finding the lock entrance during the winter upstream migration, as the higher flow of the powerplant tailwater could attract fish away from the fish lock. Although illegal, sea lampreys were frequently caught by the fisherman near the powerplant tailrace, as they appear to accumulate there (F. Godinho and P. Pinheiro, personal observations).

While the low abundance of anadromous species had influenced the absence of shads and sea lamprey in the lock, the characteristics of the Borland fish lock installed in the Crestuma-Lever could have influenced its selectivity for these species. According to [56], locks effective for shads should have main entrances situated along the banks, a surficial entrance to the lock, velocity at the entrances that is relatively high (around 2 m/s$^{-1}$), a lower chamber with dimensions compatible with shad permanence (minimum dimensions around 5 m × 2.5 m × 1.5 m), and a minimum width of the upstream chamber of around 1.5 m with a minimum flow velocity of 0.30 m$^3$/s$^{-1}$, and they should light particularly dark zones or areas with dark shadows in the lock. The Crestuma-Lever Borland fish lock lacks some of these characteristics, namely the existence of an entrance close to the riverbank, but it has several others, including its general dimensions and the amount of flow inside and at the entrance.

The lack of sea lamprey and shads in the Borland fish lock could thus have been influenced jointly by the reduced structural attractability of the fish lock and the low abundance of these species downstream of the dam.

Another aspect of selectivity relates to the number of fish using the lock in comparison with the number of fish trying to migrate upstream [57]. Although in this study no evaluation of the fish population numbers downstream of the dam was attempted, it is probable that during peak migration the number of fish trying to migrate upstream was larger than the number of fish using the lock. As an example, in the Ponte-Coimbra weir, where the fish pass operates continuously, around 500,000 upstream movements can occur annually [22]. This mismatch between the number of fish trying to migrate and using the lock derives from the discontinuous operation of the lock and cannot be changed.

*4.4. Management Implications*

Despite the limitations identified, the Crestuma-Lever fish lock was able to aid thousands of European eel and mullets passing the first dam in the Douro River. Mullets are regularly seen in the upstream Douro reservoirs, where they are able to thrive and can use the boat lock to perform some downstream displacements (F. Godinho and P. Pinheiro, personal observation). The boat lock is also used by other species for upstream displacements, including European eel, Iberian barbel and even Atlantic salmon [14]. European eels are found in some of the Douro tributaries upstream of the Crestuma-Lever dam (e.g., Arda, Paiva and Tâmega rivers), but the ecology of these individuals is still poorly known, especially if they succeed in the downstream migration. Although the presence of European eel, as the top native predator, in river fish assemblages upstream of the Crestuma-Lever dam is adequate in itself, research is needed to properly assess the fate of the European eels moving upstream through the Crestuma-Lever lock.

As for the anadromous fish species, their management in the Douro River should rely on the 20 km of the main river and its tributaries between the Crestuma-Lever dam and the Atlantic Ocean, because even if they succeeded in passing this dam, most reproducing habitats were inundated by the reservoirs. Moreover, introduced piscivorous fish that dwell in the reservoirs could reduce the success of anadromous fish moving upstream. Bearing that in mind, the use of the freshwater tributaries of the Douro estuary by anadromous species for reproduction needs to be carefully assessed, aiming at restoring and protecting the available habitat. Other protection measures could be considered, including the closing of the fishing season for shads and sea lamprey.

**Author Contributions:** Conceptualization, F.N.G. and P.J.P.; methodology, F.N.G. and P.J.P.; video analysis, F.N.G. and P.J.P., statistical analysis, F.N.G.; writing—original draft preparation, F.N.G.; writing—review and editing, F.N.G., P.J.P. and L.B. All authors have read and agreed to the published version of the manuscript.

**Funding:** This research received no external funding.

**Institutional Review Board Statement:** Not applicable.

**Data Availability Statement:** Not applicable.

**Acknowledgments:** The authors thank all the professional fishermen who gently participated in the creel survey. Thanks are also due to Alexandra Maia, Director at Munícipia, S.A. for providing the aerial image of the Crestuma-Lever dam.

**Conflicts of Interest:** The authors declare no conflict of interest.

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
