# Peer review of "Fish Use of a Borland-Type Fish Lock in an Iberian River"

_diversity, doi:10.3390/d15020178_

Round 1

Reviewer 1 Report

Main comments:

Very interesting and well-designed paper analysing the efficiency of the fish lock and I really appreciated the management proposals!

Minor comments/suggestions/corrections:

Affiliation 1, line 5, Lisboa and not “Lisbo”;

Table 1 and Table 2 – please indicate de source of the data (where determined by authors, where obtained from the dam company or elsewhere?);

Material and methods, line 113 to 121 related to non-native fish fauna: you can now include European catfish (Silurus glanis) – Martelo et al. (2021) Evaluating the range expansion of recreational non-native fishes in Portuguese freshwaters using scientific and citizen science data. BioInvasions Records (2021) Volume 10, Issue 2: 378–389. With this introduction you can alter the text from line 466 to 468 in the discussion;

Figure caption 3, line 161, right and not “wright”;

Line 177: 8h00 PM till 7h59 AM and not “e as”;

Line 178: how discharge flow was calculated?;

Line 188: Please indicate what are the “other fishing methods”;

Data analysis, ANOVA. Why did you only perform one-way ANOVA and not multiple? Some variable interaction may have occurred, as for instance between tide height and moon phase as you explain in line 399-400 at discussion.

Table 3 – refer “unidentified Cypriniforms” in last collum.

Results; Line 216: you refer 770 and 234 cycles (same in the abstract) but in table 3 is indicated a total of 754 and 218 cycles, respectively. Please verify.

Discussion, line 349: I believe it is better to use migration instead of “immigration” since is the same barbel population;

Discussion, line 381: is unknown, and not “in unknow”;

Discussion, line 460-462: Do you any idea if other species, rather than mullets, also use the boat locks?

I would like to read in the discussion a little information about the limitations of the work, if you consider that they exist. For instance, I believe that the fishing effort and the gear (large nets) used by professional fishermen were mainly directed to anadromous species, so some caution must be used when analysing the data about other species (presence or absence downstream the dam).

Author Response

Dear reviewer

We deeply appreciated the review of our manuscript. We have revise it accordingly, taking into consideration the several comments made.

The revision helped improving the ms. and we hope that it could be considered acceptable for publication in Diversity.

Please find below our reply to all the comments made.

Best regards

Francisco Nunes Godinho

Reply to comments

  • Affiliation 1, line 5, Lisboa and not “Lisbo”; - the correction was made in the ms.
  • Table 1 and Table 2 – please indicate de source of the data (where determined by authors, where obtained from the dam company or elsewhere?); - the missing information was added to the ms.
  • Material and methods, line 113 to 121 related to non-native fish fauna: you can now include European catfish (Silurus glanis) – Martelo et al. (2021) Evaluating the range expansion of recreational non-native fishes in Portuguese freshwaters using scientific and citizen science data. BioInvasions Records (2021) Volume 10, Issue 2: 378–389. With this introduction you can alter the text from line 466 to 468 in the discussion; - we have added the specific mention to Silurus glanis, and have revised the discussion to mention non-native piscivorous fish.
  • Figure caption 3, line 161, right and not “wright”; - corrected in the ms.
  • Line 177: 8h00 PM till 7h59 AM and not “e as”; - corrected in the ms.
  • Line 178: how discharge flow was calculated ? - the data were provided by the dam owner EDP that has a continuous register of the discharged flow at the powerplant.
  • Line 188: Please indicate what are the “other fishing methods”; - information added to the revised ms.
  • Data analysis, ANOVA. Why did you only perform one-way ANOVA and not multiple? Some variable interaction may have occurred, as for instance between tide height and moon phase as you explain in line 399-400 at discussion. - we recognize that some interactions could be expected. However, we were interested in the first order main effects. Moreover, given the number of factors we had considered (6 factors) it would be impractical to include them all in an ANOVA full factorial design. Nevertheless, as you pointed out, we have taken possible interactions in consideration during discussion. Moreover, we have assessed the fit of each ANOVA and have included an additional analysis in the revised ms. i.e. a general linear model (GLM), to investigate the influence of relevant (continuous and categorical) variables on the fish use of the lock. Partial η2, which measures the proportion of variance explained by a given variable of the total variance remaining after accounting for the variance explained by other variables in the model, were used to assess effect size of each variable in GLM.     
  • Table 3 – refer “unidentified Cypriniforms” in last collum. - done in the ms.
  • Results; Line 216: you refer 770 and 234 cycles (same in the abstract) but in table 3 is indicated a total of 754 and 218 cycles, respectively. Please verify. – the data were corrected in the ms (they were wrongly presented at the table).
  • Discussion, line 349: I believe it is better to use migration instead of “immigration” since is the same barbel population; - changed in the ms.
  • Discussion, line 381: is unknown, and not “in unknow”; - corrected in the ms.
  • Discussion, line 460-462: Do you any idea if other species, rather than mullets, also use the boat locks? – Other species have been found in the boat locks. - that information was added to the ms.
  • I would like to read in the discussion a little information about the limitations of the work, if you consider that they exist. For instance, I believe that the fishing effort and the gear (large nets) used by professional fishermen were mainly directed to anadromous species, so some caution must be used when analysing the data about other species (presence or absence downstream the dam) – we agree with the referee concerning the selectivity of the fishing methods used by the professional fisherman below the dam, particularly because they targeted the highly valued anadromous shads and sea lamprey. That specific aspect is already mentioned in the ms. As the main result of the comparison between fish using the lock and caught downstream is linked precisely to the anadromous taxa, we feel that the (natural) selectivity of the fishery don’t need to be highlighted.  

Reviewer 2 Report

see attachment

Author Response

Dear reviewer

We deeply appreciated the review of our manuscript. We have revise it accordingly, taking into consideration the several comments made.

The revision helped improving the ms. and we hope that it could be considered acceptable for publication in Diversity.

Please find below our reply to all the comments made.

Best regards

Francisco Nunes Godinho

Reply to comments

  • Lines 33-37: paper might flow better if this is integrated somehow into the previous paragraph. alternatively, it can probably go into the next paragraph too. Also, if you mean a Borland lock, please be specific (unless this is the only type of fish lock). I was probably initially confused because i was thinking of fish ladders, not locks. – They are all Borland, that consolidated the idea - we have rearranged the ms. text according to the suggestions.
  • Lines 59-77: how relevant is this information for the rest of your work? It sounds like maybe you are suggesting the customization and ecology of a river may strongly influence the utility of a fish lock? If so, please state that clearly. This can be also probably be a single paragraph - we are quite suggesting that. That paragraph should be interpreted after the previous one, that states in the beginning the following “Guidance on design criteria and recommendations for operation for fish locks is sparse, and as a result existing structures are somewhat unique, likely influencing their use by fish”. That uniqueness in conjunction with the variable results reported about the use of particular fish locks by fish, suggests that customization and the specific characteristics of the river (including fish species present) influence fish lock use. We believe that although implicit in the text, the idea is clear and we decided to keep it as it was. As suggested, we have merged the two paragraphs in the revised ms.
  • Line 82: Is the Douro River a 'salmonid' river? Is your point to say that the focus of much previous work has been on salmonids and not other fishes? - no, the Douro is not a “salmonid river”, except for the headwaters of some tributaries, as stated in section 2.1 of the ms. Concerning other studies conducted in fish locks in Europe, their focused particularly salmonid fishes, as did the locks built in Ireland and Scotland.
  • Lines 87-88: when was this fish lock built? Need to give a general sense of the timing since you mention dams built between 1971 and 1985 in the previous paragraph - the lock was built simultaneously with the dam (as in the other upstream dams). We have changed the text in revised ms. to clarify that.
  • Lines 90-94: is this information meant for anything else? Are you using it to increase the overall knowledge about the functioning of this lock or is there a management decision looming? - at this stage in the ms. the information is provided mainly to increase the information about the dam and the lock.
  • Line 97: Suggest making the map into Figure 1 and citing it in this sentence - we have merged Figures 1 and 2 in the revised version of the ms.
  • Lines 97-98: 'farthest downstream' might be better to say. Or do you mean it was the first dam constructed? - we fully agree with your suggestions and changed the text accordingly.
  • Figure 2: adding a small shaded watershed to the inset map will make it clear that this watershed is in northern Portugal. Also what are the other gray lines in the larger map? Are those other watershed boundaries? They either need to be defined in the legend or removed from the map - we have followed your suggestions and changed the Figure.
  • Line 122: typo…salmons - corrected in the revised ms.
  • Line 123: migrating upstream, or do these fish reside in the estuary? - they can reside, more or less temporally, in the estuary.
  • Line 132-135: move this up higher - probably before you talk about the fishes present in the river - we have done as suggested in the revised ms.
  • Line 137-140: adding a schematic to the current figure 1 and reducing the size of the satellite photo might help convey this information - following the suggestion of another referee, we have merged Figures 1 and 2. We did not include a schematic drawing of the Crestuma-Lever lock as available drawings presented a low quality and had details and characteristics in Portuguese. Nevertheless, general schemes of fish locks can be easily found on the internet (e.g. https://www.researchgate.net/publication/27335346_Fish_Locks_and_fish_lifts/figures?lo=1)
  • Line 148-150: from the results it looks like you are reporting some information on the lock cycling. I think you need to say what you recorded about the cycles, what the 'predetermined value' is, and any other information the reader needs to understanding what and why you report data in the results section. - the results presented, namely in Table 2, relate to the lock cycle characteristics, including duration of the different phases and flow and velocity in different parts of the lock. These characteristics were determined before the study begins by EDP, the owner of the hydropower dam. As stated in the ms. the lock registered a series of interventions in 2015 and 2016 to fully automate its operation, allowing the continuous succession of cycles, the stopping of cycles when the downstream levels drop below a determined value and the continuous record of each cycle stage as well as other operational details.        
  • Line 163: what is your sample size? Were all possible lock cycles included? Did You look at auto-correlation in your data? - all the lock cycles made (totalling 770) were included in the analysis. We did not look at temporal autocorrelation in the data, although some could exist as it is a phenomenon ubiquitous in ecology. We, nevertheless feel that any temporal autocorrelation in the data was not underlying any important biological process not grasped by our discussion, as it relates to fish migration and its temporal triggers.
  • Line 170: ‘counted during each cycle’…this needs more details. counted how? did you take stills from the video? - the video corresponding to the passing stage of each cycle (45’) was visualized in its entire duration. When several fishes were visualizes in the observation window, the video speed would be reduced to allow the following of each individual fish and its counting.
  • Lines 181-191: so there were multiple survey methods used downstream of the dam? If so, i would use a general statement about that as the topic sentence of the paragraph. - below the dam a single survey method was used, i.e. creel survey of the fish caught by professional fishermen with several capture methods. The global data gathered were evaluated as catch per fishing day.
  • Lines 196-201: did you run these variables individually in several ANOVAs? If no, then isn't this a multi-way ANOVA? And also if so, were you missing any variable combinations? What’s the logic for the categorization of the numerical data (both specifically and generally – e.g., why was flow split into two classes and why categorize numerical variables at all)? there may be some information lost by imposing the groups on the continuous variables. - although some of the predictors were continuous (i.e. tide height and discharged flow), others were not. That was particularly relevant for our analysis concerning sampling months, as we did not measured water temperature and used sampling month, more or less, as a surrogate for water temperature. Consequently, we opted to use in a first step categorical factors for all the explanatory variables considered and to analyse them with non-factorial ANOVA. We recognize that some interactions could be expected among the explanatory variables considered, but the number of factors we had considered (6 factors) would make it impractical to include them all in a full ANOVA factorial design. Nevertheless, we have taken possible interactions in consideration during discussion. Moreover, in the revised ms. we have assessed the fit of each ANOVA and have included an additional analysis. General linear models (GLM) were performed to investigate the influence of relevant (continuous and categorical) variables on the fish use of the lock. Partial η2, which measures the proportion of variance explained by a given variable of the total variance remaining after accounting for the variance explained by other variables in the model, were used to assess effect size of each variable in the GLM.
  • Line 216: ‘with fish’…at least one fish? Was there ever just one fish? ­– We have used fish for the plural.
  • Lines 218-219: exclude words like 'rather' Occurrence of fish per lock cycle was lowest in March and April and highest when? This data is shown in Table 3? - the ms. was revised accordingly to your suggestion. All information presented in the text from the results section can be simply derived from the data provided in Table 3. 
  • Lines 229-230: you may want to add more detailed data tables to the supplemental information. Table 3. - all the relevant information on fish use of the lock is already presented in Table 3, including, for each month, the total number of cycles, the number of cycles with fish and the number of each taxa visualized.
  • Lines 231-235: how did you address forced perspective? (A fish closer to the viewing window appears larger) - as reported in ms., lengths of some of the fishes were approximately measured, mainly to get an idea of the maximum sizes using the lock. To reduce errors, we mostly measured fish close to the viewing window that has a known length (0.595 cm), although the reticulate in the back panel was also used. Note also that the width of the viewing chamber is reduced compared to the upstream chamber, not only to facilitate the observation and the counting of fish, but also to reduce measuring errors linked with forced perspective.
  • Line 240: what is ‘ind.cycle-1’? This needs to be defined in the caption - it is defined in the caption in the revised version of the ms.
  • Line 258: average is the mean here? Remember to be specific - changed in the revised ms.
  • Line 260: typo…Creed survey - changed in the revised ms.
  • Line 262: how long is a ‘fishing day’? Is it defined in the methods? If not, it needs to be - a fishing day usually encompasses fishing throughout the night, including repeatedly setting and lifting the nets. The exact duration can vary according to the fishing success and the discharged flow at the powerplant tailwater. That information was added to the ms.
  • Lines 269-274: were any species found in one survey (creel vs lock) but not the other? - For example, all the anadromous species caught by the fishermen were never found in the lock during the same period. Those results are described in section 3.3. of the ms.
  • Lines 275-282: what about the netting you mentioned in the methods? - we don’t fully understand the comment. Eel was caught by the fishermen with different fishing gear, but more importantly, the eel catch was not presented in Figure 6 (original ms.) because it was reported in weight and not in numbers.
  • Line 286: state the actual year; also remind the reader somewhere that the periods of the fishing surveys and the fish lock study did not overlap exactly. Arguably, it'd make sense to drop the non-corresponding data (if one of your questions is to look at the relationships between fishing and fish migrations/function of the lock ­­– please note that the entire period that the lock was studied (November 2016 – July 2017) was comprehended in the period when the creel survey was made (August 2016 -July 2017). We kept the creel survey data for the entire year as we believe it could be useful for future reference.
  • Line 288: define the coding for the variables in the caption - changed in the revised ms.
  • Table 5: define the species captured in this 'Others' category in the table caption. - changed in the revised ms.
  • Line 302-308: this paragraph probably needs a topic sentence – we have merged the two first paragraphs of the discussion and feel that the reading is improved.
  • Lines 304-306: in the previous work? - as stated in that paragraph, a preliminary study has been conducted at the same fish lock from the end of October 1994 to the middle of March 1995.
  • Line 308: typo - corrected in the revised ms.
  • Lines 309-319: so how does this relate to your work? What is the major point you are trying to make here? Seasonality matters? Fish locks get used more over time? - a topic phrase was added at the beginning
  • Lines 320-329: these paragraphs are disjointed and there are results sprinkled in the discussion - we don’t fully agree with the disjunction. The first paragraph links the higher fish movements (and thus higher potential use of the lock) to migrations and not to dispersal (i.e. one-way movement, away from a site as a result of individual behavioural decisions made at different life stages, temporal and spatial scales individual movements), concluding by stating that the taxa using the lock more intensely are indeed migratory species (catadromous eel and mugilids). The second paragraph relates the findings from other studies that the increased movement (and thus potential use of the lock) during migration has been linked to the increase in water temperature occurring during Spring and early Summer. Note that we did not measured water temperature and consequently sampling months can be interpreted as a surrogate for temperature, as temperature increases continuously from Winter to Summer at the studied latitudes. We have changed the second paragraph to make it clearer, by replacing higher with increasing.
  • Lines 331-338: again, what is the main point you are making here? The use of the fish locks match known life cycles and environmental cueing? - If the lock was completely non-selective for particular taxa, the same factors naturally influencing fish upstream movement would be the ones influencing fish lock use, and consequently, the use should match known life cycles and environmental cueing.
  • Lines 347-349: these are results - you are right and the phrase was deleted. Its mains idea was added to the results.
  • Section 4.2: the discussion of the influence of temperature probably belongs in this section. Did you measure water temperature? - as stated before we did not measured temperature and have used the sampling month as a surrogate for its influence on lock use.
  • Line 372: “the proximate attractability of fish passes”…this can probably be stated more plainly - we agree and have deleted proximate
  • Lines 383-384: well you really looked at tide height categories - Nevertheless, if the fish used the lock with tide height > 2 m in a location where average tidal range is 2.8 m (information added to the ms. - section 2.2. study area), that can be considered a higher tide.
  • Line 405: what two lock cycles tested? - as stated in the end of section 2.2., two different lock cycles were assessed, either including or not the flow generated by the lateral outlets. The outlets are described in section 2.1 of the ms.
  • Line 412-415: was this described specifically in your results? If not it needs to be. - it was described in section 3.3. of the ms., where it is stated that “When comparing the fish caught downstream the Crestuma-Lever dam and the fish visualized in the lock the most striking difference related to the anadromous shad and sea lamprey, that despite being captured by fisherman from December until May were never found in the fish lock”
  • Lines 415-417: this is a good topic sentence for the next paragraph – suggest moving it down –
  • Line 422: what is the red data book? Please be specific. - we have changed the reference from the Portuguese Red data book to the Portuguese Red list of endangered fish species.
  • Lines 423-434: this could probably be a new paragraph - done in the revised ms.
  • Line 470: 'last' is a relative qualifier - do you mean the 20 km of the river between the lock and the Atlantic Ocean? - yes, we mean that. We have changed the phrase.
  • Lines 471-475: similar to my comment in the intro, is this an objective of the paper? Are you providing official advice to decision makers with this paper? Based on results – yes, we are (unofficially) suggesting the need to assess the use of the freshwater tributaries of the Douro estuary by anadromous species for reproduction, as there are likely the remnant areas available for the populations of these species to survive in the Douro River.

Reviewer 3 Report

This MS analyses the use of Borland fish lock installed in the Crestuma-Lever dam (Douro River, Portugal) by different fish species during 9 months, assessing the temporal variation in fish use of the lock, identifying the environmental triggers of the lock use by different species, and analysing the selectivity in the use of the lock by different fish species.

The MS is well written and organized. Results are clearly present and the discussion is supported by the presented results.

I only have minor comments for the Authors.

Line 13-14: Please confirm the number of lock cycles and cycles with fish presented in the Abstract.

Line 30: replace “fish locks are” with “fish locks were”

Line 97: replace “The fish lock investigated” with “The studied fish lock”

Table 1: please confirm if the units are written correctly and if s-1 is really superscript

Line 116: replace “taxa” with “species”

Lines 151-152: How these operational details are measured? I suggest the Authors give more detailed information about it in the text.

Lines 173-174: Why were the fish not measured using an image analysis program from the frames of the video files? Are the produced fish lengths acceptably accurate for this study?

Line 216: Values presented in the text related to the total number of cycles (770) and cycles with fish (234) are different from those presented in Table 3 (754 and 218, respectively).

Line 221: replace “taxa” with “species”

Line 222: replace “taxa” with “fish species”

Lines 237-238: The Authors analyzed each variable that could be related to the number of fish using fish locks separately. But isn't there any interaction between the variables that could explain the number of fish that use the locks? Wouldn't it be more informative to present a model that defines the number of fish that uses the fish lock as a function of each of the significant explanatory variables?

Line 240 (Table 4 caption): Please superscript -1 in "cycle-1".

Line 246 and 247: Please superscript -1 in "cycle-1".

Line 257: replace “taxa” with “fish species”

Line 258: replace “average” with “mean”, and “investigated” with “analysed”

Line 269: replace “taxa” with “fish species”

Figure 6: I suggest authors change the type of graph displayed, as it gets a bit confusing. A 100% stacked column chart seems more suitable in this case. In addition, on the x-axis, the corresponding year of each month should be added.

Lines 357-360: The authors should explain why both larger flows and higher flows act as triggers for fish migration for the same species.

Line 390: replace “taxa” with “fish species”

Line 411: replace “taxa visualized” with “studied fish species”

Line 414: replace “taxa” with “species”

Line 423: replace “taxa” with “species”

Line 426: replace “taxa” with “species”

Line 435: replace “taxa” with “species”

Line 437: replace “taxa” with “species”

Line 448: replace “taxa” with “species”

Line 469: replace “taxa” with “fish species”

Line 475: add a “.” at the end of the sentence.

Author Response

Dear reviewer

We deeply appreciated the review of our manuscript. We have revise it accordingly, taking into consideration the several comments made.

The revision helped improving the ms. and we hope that it could be considered acceptable for publication in Diversity.

Please find below our reply to all the comments made.

Best regards

Francisco Nunes Godinho

Reply to comments

  • Line 13-14: Please confirm the number of lock cycles and cycles with fish presented in the Abstract. - corrected in the ms (the numbers were wrong in Table 3).
  • Line 30: replace “fish locks are” with “fish locks were” - replaced in the revised ms.
  • Line 97: replace “The fish lock investigated” with “The studied fish lock” - replaced in the revised ms.
  • Table 1: please confirm if the units are written correctly and if s-1 is really superscript - corrected in the ms.
  • Line 116: replace “taxa” with “species” - replaced in the revised ms.
  • Lines 151-152: How these operational details are measured? I suggest the Authors give more detailed information about it in the text. - additional information was added to the ms.
  • Lines 173-174: Why were the fish not measured using an image analysis program from the frames of the video files? Are the produced fish lengths acceptably accurate for this study? - as reported in ms., lengths of some of the fishes were approximately measured, mainly to get an idea of the maximum sizes using the lock. To reduce errors, we mostly measured fish close to the viewing window, that has a known length (0.595 cm), although the reticulate in the back panel was also used. Note also that the width of the viewing chamber is reduced compared to the upstream chamber, not only to facilitate the observation and the counting of fish, but also to reduce measuring errors linked with forced perspective.
  • Line 216: Values presented in the text related to the total number of cycles (770) and cycles with fish (234) are different from those presented in Table 3 (754 and 218, respectively). - corrected in the revised ms (the numbers were wrong in Table 3).
  • Line 221: replace “taxa” with “species” - replaced in the ms.
  • Line 222: replace “taxa” with “fish species” - replaced in the ms.
  • Lines 237-238: The Authors analyzed each variable that could be related to the number of fish using fish locks separately. But isn't there any interaction between the variables that could explain the number of fish that use the locks? Wouldn't it be more informative to present a model that defines the number of fish that uses the fish lock as a function of each of the significant explanatory variables? - we recognize that some interactions could be expected among factors. However, we were interested in the first order main effects from the several factors considered. Moreover, given the number of factors we had considered (6 factors), it would be impractical to include them all in a full ANOVA factorial design. Nevertheless, as you pointed out, we have taken possible interactions in consideration during discussion. Moreover, we have assessed the fit of each ANOVA and have included an additional analysis in the revised ms. i.e. general linear models (GLM), to investigate the influence of relevant (continuous and categorical) variables on the fish use of the lock. Partial η2, which measures the proportion of variance explained by a given variable of the total variance remaining after accounting for the variance explained by other variables in the model, were used to assess effect size of each variable in the GLM.
  • Line 240 (Table 4 caption): Please superscript -1 in "cycle-1". - corrected in the revised ms.
  • Line 246 and 247: Please superscript -1 in "cycle-1". - corrected in the revised ms.
  • Line 257: replace “taxa” with “fish species” - as one of the lines in fact represents a species group, taxa was kept.
  • Line 258: replace “average” with “mean”, and “investigated” with “analysed” - replaced in the ms.
  • Line 269: replace “taxa” with “fish species” - replaced in the revised ms.
  • Figure 6: I suggest authors change the type of graph displayed, as it gets a bit confusing. – the A 100% stacked column chart seems more suitable in this case. In addition, on the x-axis, the corresponding year of each month should be added. - the figure was changed according to the suggestions.
  • Lines 357-360: The authors should explain why both larger flows and higher flows act as triggers for fish migration for the same species - that reference was indeed a bit confusing and we have deleted it from the ms.
  • Line 390: replace “taxa” with “fish species” - replaced in the ms.
  • Line 411: replace “taxa visualized” with “studied fish species” - replaced in the ms.
  • Line 414: replace “taxa” with “species” - replaced in the ms.
  • Line 423: replace “taxa” with “species” - replaced in the ms.
  • Line 426: replace “taxa” with “species” - replaced in the ms.
  • Line 435: replace “taxa” with “species” - replaced in the ms.
  • Line 437: replace “taxa” with “species” - replaced in the ms.
  • Line 448: replace “taxa” with “species” - replaced in the ms.
  • Line 469: replace “taxa” with “fish species” - replaced in the ms.
  • Line 475: add a “.” at the end of the sentence. - corrected in the ms.

Reviewer 4 Report

The manuscript presents the performance of the Borland type fish lock in the Douro River, Portugal and link to the factors that trigger fish migration. Results of the study can be case study and lesson learn on the effort to conserve the fish diversity by the impact of dam development. I have made some comments for improving this MS as below;

Major

·      Too many short paragraphs, suggest merging paragraphs #2 & #3; #5 & #6 and #7 & # 8. Or paragraph #5, #6, #7 and # 8 together

·      Better to combine Figure 1 and 2

·      Why don’t the logbook was used for creel survey but interview? Please elaborate

·      The posteriori-test, e.g. Duncan’s multiple range test, must be followed once the ANOVA results showed significant difference (P < 0.05) for Month and Moon phase

·      Explanation from Lines 243-255 is very confused, why “-” was used as symbol?

·      Influence from each different trigger must be not clear presented, i.e. which trigger is likely the most influence factor?

·      Text and graphical results showing the fluctuation in catches of European eel must be presented. Also, what is the unit for catches of European eel? Total catches from creel survey? Catch from individual fisher?

·      Results in sub-section 3.3 is not clear and must be revised. Also, Discussion in #4.3 is seem exaggerated, comparing to the Results the authors presented in #3.3

·      Discussion better to relate the biology of the diadromous species found in the study.

·      The is no clear discussion, which link the results from monitoring and creel surveys. Would fishing intensity effect the number of fish migration? This should be address also in Conclusion.

Minor

·      Line 90 and elsewhere in text: 1970’s and 1980’s

·      If possible, Paragraph 2 of the introduction better drawing picture

·      Not understand why “taxa” is italic?

·      Introduction, last paragraph: Please elaborate “nine months” , when was it exactly?

·      Check unit in Table 1

·      Lines 134-135: Please use “psu” for salinity unit

·      Using of the unit in Table 4, please make full not abbreviation

·      Line 264: The correlation was made during which period, please elaborate.

END OF REVIEW

Author Response

Dear reviewer

We deeply appreciated the review of our manuscript. We have revise it accordingly, taking into consideration the several comments made.

The revision clearly helped improving the ms. and we hope that it could be considered acceptable for publication in Diversity.

Please find below our reply to all the comments made.

Best regards

Francisco Nunes Godinho

Reply to comments

  • Too many short paragraphs, suggest merging paragraphs #2 & #3; #5 & #6 and #7 & # 8. Or paragraph #5, #6, #7 and # 8 together - changed in the revised ms. The paragraphs #2 & #3; #5 were merged.
  • Better to combine Figure 1 and 2 - as suggested, the two figures were combined.
  • Why don’t the logbook was used for creel survey but interview? Please elaborate - the fishery downstream the Crestuma-Lever dam has a special regulation and fisherman don’t have logbooks.
  • The posteriori-test, e.g. Duncan’s multiple range test, must be followed once the ANOVA results showed significant difference (P < 0.05) for Month and Moon phase – as suggested we have included the results of Duncan Post Hoc tests.
  • Explanation from Lines 243-255 is very confused, why “-” was used as symbol? - we have made some changes to the text that have hopefully clarify its content.
  • Influence from each different trigger must be not clear presented, i.e. which trigger is likely the most influence factor? - in the revised ms. we have assessed the fit of each ANOVA and have included an additional analysis, i.e. general linear model (GLM), to investigate the influence of relevant (continuous and categorical) variables on the fish use of the lock. Partial η2, which measures the proportion of variance explained by a given variable of the total variance remaining after accounting for the variance explained by other variables in the model, were used to assess effect size of each variable in the GLM.
  • Text and graphical results showing the fluctuation in catches of European eel must be presented. Also, what is the unit for catches of European eel? Total catches from creel survey? Catch from individual fisher? - as reported in the Figure caption the data presented in Figure 6 (original submission) are (for each studied month) the proportion of the total number of fish caught by fishermen represented by the different species caught downstream the Crestuma-Lever dam. We have changed the caption to clarify the Figure data. The fluctuation in European eel catch, as well as the fluctuations in the quantities of the other taxa caught downstream the dam by fisherman are presented in Table 5. We did not made any Figure with those data, as it would basically repeat the information already presented.
  • Results in sub-section 3.3 is not clear and must be revised. Also, Discussion in #4.3 is seem exaggerated, comparing to the Results the authors presented in #3.3 - section 3.3 simply highlights the striking difference observed between the fish species using the lock and the fish caught downstream the dam, that is related to the anadromous taxa (shads and sea lamprey). These species, despite being caught below the dam by the professional fisherman were never found in the lock. We have added some text to increase readability. Although the results are somewhat presented with a short description in section 3.3., the discussion encompasses several important aspects related to the selectivity of the fish lock that we believe are important to keep in the ms.
  • There is no clear discussion, which link the results from monitoring and creel surveys. Would fishing intensity effect the number of fish migration? This should be address also in Conclusion. - the survey of the fish caught downstream the dam was made to get an idea of the fish assemblage present, i.e. the species that could potentially use the fish lock. Consequently, the creel survey was not the main focus of the study and we were not interest in other aspects, namely the influence of fishing intensity on the number of fish using the lock. Nevertheless, given the reduced fishing effort, it is safe to say that, except for anadromous taxa, fishing done by fisherman did not influence the number of fish using the lock.
  • Line 90 and elsewhere in text: 1970’s and 1980’s - corrected in the ms.
  • If possible, Paragraph 2 of the introduction better drawing picture - Figures 1 and 2 were merged. However, we didn´t presented any scheme of the Crestuma-Lever fish lock, as available drawings were old and lacked quality. Nevertheless, general schemes of fish locks can be found easily on the internet (e.g. https://www.researchgate.net/publication/27335346_Fish_Locks_and_fish_lifts/figures?lo=1)
  • Not understand why “taxa” is italic? - we have replaced taxa with species for most parts in the ms. The remaining mentions are no longer italicized.
  • Introduction, last paragraph: Please elaborate “nine months”, when was it exactly? - the details about the months studied are given in section 2.3. of the ms. (“The monitoring of the fish lock began in November 2016 and ended in July 2017).
  • Check unit in Table 1. -
  • Lines 134-135: Please use “psu” for salinity unit - we have used the suggested unit designation.
  • Using of the unit in Table 4, please make full not abbreviation - done in the revised ms.
  • Line 264: The correlation was made during which period, please elaborate. - we have added some text in the paragraph, as well as in Figure 5 (of the original ms.), to make the idea clearer.

Round 2

Reviewer 2 Report

Thank you for responding to the previous review. I have no further comments.